# Bioinformatics Approach for Identifying Novel Biomarkers and Their Signaling Pathways Involved in Interstitial Cystitis/Bladder Pain Syndrome with Hunner Lesion

**DOI:** 10.3390/jcm9061935

**Published:** 2020-06-21

**Authors:** Subbroto Kumar Saha, Tak-Il Jeon, Soo Bin Jang, Se Jong Kim, Kyung Min Lim, Yu Jin Choi, Hyeong Gon Kim, Aram Kim, Ssang-Goo Cho

**Affiliations:** 1Department of Stem Cell and Regenerative Biotechnology, Molecular & Cellular Reprogramming Center (MCRC), and Incurable Disease Animal Model & Stem Cell Institute (IDASI), Konkuk University, 120 Neungdong-ro, Gwangjin-gu, Seoul 05029, Korea; subbroto@konkuk.ac.kr (S.K.S.); jeonti94@naver.com (T.-I.J.); buruburu6586@naver.com (S.B.J.); rlatpwhdc@nate.com (S.J.K.); lmin0217@naver.com (K.M.L.); trikk33@naver.com (Y.J.C.); 2Department of Urology, Konkuk University Medical Center, Konkuk University School of Medicine, Seoul 05029, Korea; 20040097@kuh.ac.kr

**Keywords:** interstitial cystitis, bladder pain syndrome, diagnosis, biomarkers, bioinformatics, signaling pathway

## Abstract

The complexity of interstitial cystitis/bladder pain syndrome (IC/BPS) has led to considerable uncertainty in terms of diagnosis and prevalence of the condition. Here, we try to identify the IC/BPS-associated genes through an integrated analysis of Gene Expression Omnibus (GEO) datasets and confirm experimentally to predict the pathologic diagnosis of IC/BPS. Data mining analysis of GEO datasets (GSE621, GSE11783, GSE28242, and GSE57560) revealed a total of 53 (51 upregulated and two downregulated) common differentially expressed genes (DEGs) in IC/BPS. A protein–protein interaction (PPI) network was then constructed with the 53 common DEGs using Cytoscape v3.7.2, and subsequently, six hub genes (*CD5*, *CD38*, *ITGAL*, *IL7R*, *KLRB1*, and *IL7R*) were identified using cytoHubba v0.1 that were upregulated in IC/BPS. Enrichment analysis of common DEGs revealed that hematopoietic cell lineage, immune system, and T-cell receptor (TCR) signaling in naïve CD4+ T cell signaling pathways were prominently involved with the common 51 upregulated DEGs. The two common downregulated DEGs may enrich linoleic acid metabolism and synthesis of epoxy (EET) and dihydroxyeicosatrienoic acid (DHET) signaling pathways in IC/BPS. Moreover, our RT-PCR data confirmed that the expression of the five hub genes (*CD38*, *ITGAL*, *IL7R*, *KLRB1*, and *IL7R*) was significantly augmented in IC/BPS patients’ samples when compared with their normal counterparts. In this study, we systematically predict the significant biomarkers and possible signaling pathways involved in IC/BPS, confirming the differential expression of the hub genes in tissue samples from patients with IC/BPS. Thus, the hub genes might be used as potential diagnostic biomarkers of IC/BPS.

## 1. Introduction

Interstitial cystitis/bladder pain syndrome (IC/BPS) is a prevalent condition that presents with symptoms of disrupted urine storage and pelvic pain. The syndrome lacks definitive diagnostic tests or markers and is defined by a collection of symptoms reported by patients, making diagnosis subjective and clinician dependent.

Overactive bladder (OAB) also exhibits storage symptoms without pelvic pain. Moreover, there is no biological test to diagnose the disease. IC/BPS is characterized predominantly by bladder pain, including the subjective sensations of ‘pressure’ and ‘discomfort,’ whereas the hallmark symptom of OAB is urinary urgency. The presence of bladder pain in OAB or urinary urgency/urgency incontinence in IC/BPS has been considered rare, but recent evidence suggests considerable symptom overlap between IC/BPS and OAB [1,2,3,4]. While there are several diagnostic tests used for IC/BPS, such as a urinalysis, urine culture, potassium sensitivity testing, cystoscopy, and biopsy of the bladder, none of these tests can definitively diagnose IC/BPS. Since there are no specific diagnostic criteria for IC/BPS, the diagnosis is most often made when other conditions have been excluded [1]. While pain related to the bladder is the hallmark symptom of IC/BPS, there is no single definition of IC/BPS that can both identify all IC/BPS cases and distinguish these cases from similar conditions, such as OAB, endometriosis, and vulvodynia [5]. Under these circumstances, chronic inflammation has been reported to have arisen after surgery in IC/BPS patients, which is frequently misdiagnosed. The purpose of this study is to assess the potential of a genetic diagnosis of IC/BPS. We conducted an extensive bioinformatic analysis to identify genes and transcriptional factors to determine whether these biomarkers meet the clinical criteria to confirm the diagnosis of IC/BPS.

## 2. Experimental Section

### 2.1. Identification of Differentially Expressed Genes (DEGs) from Gene Expression Omnibus (GEO) Datasets of IC/BPS

To identify DEGs in IC/BPS, we utilized the Gene Expression Omnibus (GEO), National Center for Biotechnology Information (NCBI), and National Institutes of Health (NIH), https://www.ncbi.nlm.nih.gov/geo/) and searched data of interstitial cystitis/bladder pain syndrome (IC/BPS). Four datasets were used in our analyses, and these included: GSE621, GSE11783, GSE28242, and GSE57560. The dataset GSE621 contained human cDNA gene array analysis of cell lines treated with/without antiproliferative factor (APF) (*n* = 16), IC/BPS (*n* = 6), and normal (*n* = 6) tissue samples using GPL262 platform. In our study, we included DEGs only from IC/BPS vs. normal tissue samples. It is noted that IC/BPS tissues were not classified as Hunner lesions (HL) and non-HL [6]. The dataset GSE11783 contained the gene expression profiles of bladder biopsy tissues from IC/BPS (*n* = 10) and control (*n* = 6) patients using Affymetrix Human Genome U133 Plus 2.0 Array, GPL570 platform. The IC/BPS tissues were also classified by patients with ulcer and non-ulcer [7]. The dataset GSE57560 contained the gene expression profiles of cystoscopic bladder biopsy tissues from IC/BPS (*n* = 8) and control (*n* = 8) patients using Agilent-039494 SurePrint G3 Human GE v2 8x60K Microarray 039381, GPL16699 platform [8]. The dataset GSE28242 contained gene expression profiles of urine sediment from IC/BPS (*n* = 8) and control (*n* = 5) patients using Affymetrix Human Gene 1.0 ST Array, GLP6244 platform. The IC/BPS tissues were also classified into patients with HL and non-HL [9]. In this study, we only included patients from normal and IC/BPS samples with Hunner lesions for each dataset (Appendix A). DEGs were retrieved from datasets comparing IC/BPS and normal tissues. To present significant DEGs with a higher fold change, we prepared volcano plots using iLINCS (Integrative LINCS, an integrative web platform for the analysis of LINCS data and signatures, LINCS DCIC, http://ilincs.org) [10]. DEGs in IC/BPS that met a *p*-value threshold < 0.05 and logarithmic fold change, log2FC < 2 or log2FC > −2 were considered significant.

To identify common DEGs among GEO datasets, we drew a Venn diagram using upregulated and downregulated DEGs separately acquired from the three GEO datasets, including GSE11783, GSE28242, and GSE57560. Subsequently, the expression of common DEGs, which were retrieved from the Venn diagram, was presented as a heatmap of hierarchical clustering analysis using the MeV software v4.9.0 (Multiple Experiment Viewer). The color of the pattern depends on the gene expression level with the threshold of −1 < z-score < 1. High and low values outside the criteria were assigned as the highest and lowest values.

### 2.2. Gene Ontology (GO) and Pathway Analysis of Common DEGs

The putative pathway and Gene Ontology of the common upregulated and downregulated DEGs were analyzed using Enrichr (Mount Sinai Innovation Partners, https://amp.pharm.mssm.edu/Enrichr/), a web server for comprehensive gene set enrichment analysis [11,12]. In addition, the expected pathways and GO of hub genes were also analyzed through Enrichr. The bar graphs retrieved from Enrichr were ranked by *p*-value from several databases, including the Kyoto Encyclopedia of Genes and Genomes (KEGG), Reactome pathway, National Cancer Institute (NCI)-Nature Pathway, and GO (biological process, molecular function, and cellular component).

### 2.3. Protein-Protein Interaction (PPI) Construction and Identification of Hub Genes

To construct the PPI network, we utilized the Cytoscape v3.7.2 software (NRNB, NIH/NIGMS/BBCB) with the help of the STRING application (©STRING consortium 2019, ELIXIR, https://string-db.org/) using common DEGs [13,14]. This tool can provide interactions between intersected genes. Subsequently, we used cytoHubba through the Cytoscape v3.7.2 software for finding the top six hub genes, which might be interactive biomarkers of IC/BPS [15,16].

### 2.4. Construction of Hub Genes and the Transcriptional Factor (TF) Network

To construct a hub gene and TF network, we utilized the web-based tool NetworkAnalyst (http://www.networkanalyst.ca/faces/home.xhtml) [17,18]. It can visualize the network of hub gene and TF interactions for input genes. In this study, the hub genes and their predicted regulatory TF networks were constructed through NetworkAnalyst using the JASPAR TF binding site profile database and visualized using the Cytoscape v3.7.2 software.

### 2.5. Clinical Sample Collection from IC/BPS and Normal Bladder Tissues

This prospective study was approved by the Institutional Review Board of Konkuk University Medical Center (IRB approval number: KUH1130060, KUMC 2019-07-009), and the present research was performed in accordance with the Declaration of Helsinki. For enrollment in the study, we obtained informed consent from all patients.

We strictly followed the American Urological Association criteria to diagnose IC/BPS [1]. Assessments of patient symptoms included the Visual Analogue Score pain questionnaire (VAS), O’Leary–Sant Interstitial Cystitis Symptom Index (ICSI) and O’Leary–Sant Interstitial Cystitis Problem Index (ICPI) (IC-Q), and Pelvic Pain and Urgency/Frequency Patient Symptom/Bother score (PUF). We enrolled patients with >13 points on the PUF, >24 points on the IC-Q, and >4 points on the VAS. Cystoscopy, urine culture, cytology, and physical examination were performed to categorize the subtypes of IC/BPS and to exclude cases with urinary tract infections, bladder cancer, urinary tuberculosis, urolithiasis, other neurological diseases, or endometriosis.

Based on cystoscopic findings, we confirmed that the patients had Hunner lesions. Regarding treatment, hydrodistension, and transurethral resection and coagulation (TUR-C) for Hunner lesions were performed. The IC/BPS patient-derived specimens (IC/BPS 1–7) were obtained from Hunner lesions during TUR-C. Due to performing TUR, mucosal and submucosal layers were included in the tissues from Hunner lesions. In the case of control specimens (non-IC/BPS 1–5), the tissues were collected from the bladders when the patient underwent radical cystectomy (non-IC/BPS 1) or when the patient transurethral resection of bladder tumors (non-IC/BPS 2–5). The tissues away from the tumor sites were carefully sampled and these tissues were confirmed to be non-malignant. All non-IC patients showed no lower abdominal pain when the bladder was filled with urine and no lower urinary tract symptoms. For selecting the patients, the inclusion and exclusion criteria of the study population are displayed in Table 1. The bladder tissue samples obtained from patients were immediately processed for examination. After collection, tissues were stored at −80 °C for RNA extraction. The methodology used for processing these samples is described in the next section. 

### 2.6. Real-Time Quantitative Polymerase Chain Reaction (RT-qPCR) Analyses

LaboPass™ Labozol reagent (Cosmo Genetech, CMRZ001, Seoul, Korea) was used to isolate total RNA from clinical tissues. Afterward, extracted total RNA was reverse-transcribed into cDNA (complementary DNA) by utilizing a LaboPass™ M-MuLV Reverse Transcriptase kit (Cosmo Genetech, CMRT010, Seoul, Korea). HiPi Real-Time PCR 2x Master Mix (ELPISBIO, EBT-1802, Daejeon, Korea) was used to measure the relative expression level of 6 hub genes between control and IC/BPS patient tissues. The primers used in this study are listed in Table 2. The respective methods were performed according to the manufacturer’s protocol. The relative expression results were normalized using *GAPDH* expression, an internal control, and expression fold change of a gene was calculated using the comparative ΔCT (cycle time) method.

### 2.7. Statistical Analysis

Data analysis was performed using GraphPad Prism version 7 (GraphPad Software, La Jolla, CA, USA). Hub gene data from GEO datasets were made into respective histograms and were analyzed for statistical significance. These results were compared between the IC/BPS and normal group, and statistical significance was calculated using a two-tailed unpaired *t*-test. A *p*-value < 0.05 was considered statistically significant (* *p* < 0.05, ** *p* < 0.01, *** *p* < 0.001, **** *p* < 0.0001).

## 3. Results

### 3.1. Identification of DEGs from IC/BPS Patient Tissues

In this study, we retrieved GEO datasets from microarray analyses of IC/BPS with Hunner lesions using the GEO database. Then, we identified several DEGs from the four GEO datasets via DEG analysis using the LINCS DCIC. These included 184 genes in GSE28242, 6756 genes in GSE57560, 0 genes in GSE621, and 4702 genes in GSE11783 datasets (Figure 1a–d). As we did not find any DEG that aligned with our criteria, the GSE621 dataset was excluded from further analyses (Figure 1a). Among the DEGs from the three GEO datasets, we separately obtained 51 overlapping upregulated (Figure 2a) and two downregulated (Figure 2b) DEGs through a Venn diagram. All the DEGs are listed in Appendix A. Expression heatmaps were also drawn for overlapped DEGs using the expression data from GSE28242, GSE57560, and GSE11783 datasets (Figure 3a–c). These maps reveal that the expression of selected genes generally was significantly different in IC/BPS vs. the control group.

### 3.2. Enrichment Analyses of DEGs from IC/BPS Patient Tissues

We then performed enrichment analyses for upregulated and downregulated DEGs using Enrichr. Based on our analysis, we found various predominant pathways, particularly associated with upregulated DEGs. These included hematopoietic cell lineage, viral myocarditis, and *Staphylococcus aureus* infection in the KEGG 2019 pathway (Figure 4ai); immune system, adaptive immune system, and cytokine signaling in the immune system from the Reactome pathway 2016 (Figure 4aii); and T-cell receptor (TCR) signaling in naïve CD4+ T cells, integrin family cell-surface interactions, and CXCR4-mediated signaling events from the NCI-Nature Pathway 2016 (Figure 4aiii). Several pathways were also associated with downregulated DEGs. For example, linoleic acid metabolism, ovarian steroidogenesis, and arachidonic acid metabolism were pathways predominantly enriched in the KEGG 2019 pathway (Figure 4bi); synthesis of epoxy (EET) and dihydroxyeicosatrienoic acid (DHET), fatty acid, and xenobiotics, and Reactome pathway 2016 may be associated with downregulated DEGs (Figure 4bii).

Next, we performed Gene Ontology (GO) analysis using common upregulated and downregulated DEGs via Enrichr. The uppermost GO terms (biological process, molecular function, and cellular component) are presented in Figure 4. In terms of biological processes, the common upregulated DEGs were most significantly associated with antigen receptor-mediated signaling pathways, whereas linoleic acid metabolic process was the most enriched GO term for common downregulated DEGs (Figure 4aiv,biii). Our subsequent molecular function enrichment analysis showed that the major histocompatibility complex (MHC) class II protein complex was the most general term associated with upregulated DEGs, while downregulated DEGs were related to arachidonic acid monooxygenase activity (Figure 4av,biv). For cellular components, upregulated DEGs were most significantly enriched in MHC class II protein complex binding, but downregulated DEGs were not enriched in any cellular component (Figure 4avi).

### 3.3. PPI Network Analyses of DEGs and Hub Gene Identification from IC/BPS Patient Tissues

A PPI network was constructed with common upregulated and downregulated DEGs by using the Cytoscape v3.7.2 software with the help of the STRING plugins. As shown in Figure 5a, there were 53 nodes (51 upregulated DEGs and two downregulated DEGs) and 47 edges in the network. Subsequently, the top six hub genes were identified from the network using the cytoHubba plugins through the Cytoscape v3.7.2 software (Figure 5b). These hub genes were cluster of differentiation 5 (*CD5*), cluster of differentiation 38 (*CD38*), integrin alpha L chain (*ITGAL*), interleukin 7 receptor (*IL7R*), killer cell lectin-like receptor B1 (*KLRB1*), and proteasome subunit beta 9 (*PSMB9*).

We next constructed a hub gene–TF network through NetworkAnalyst using JASPAR database, the TF binding site profile database. From the hub gene–TF network, we found that six hub genes interacted with 26 regulatory TFs (Figure 5c). Among the hub genes, *ITGAL* was the most prominent gene regulated by 13 TFs; *PSMB9* was regulated by eight TFs; *CD5* was regulated by seven TFs; *CD38* and *IL7R* were regulated by five TFs; and *KLRB1* was regulated by four TFs (Figure 5c, left panel). Moreover, several TFs were found to associate with more than one hub gene. Specifically, 10 TFs were observed in the network to interact with ≥2 hub genes (Figure 5c, right panel), which implies that these TFs might closely interact with the mentioned hub genes. For example, GATA2 was found to regulate *CD5*, *CD38*, *ITGAL*, *IL7R*, and *KLRB1*, while YY1 may regulate *ITGAL*, *KLRB1*, and *PSMB9*.

### 3.4. Enrichment Analyses of Hub Genes from IC/BPS Patient Tissues

To identify specific pathways and GO enrichment of biomarkers, we again performed enrichment analysis using only hub genes via Enrichr. The top enriched pathways and GO terms are shown in Figure 6. In terms of pathway analysis, hub genes were predominantly enriched in hematopoietic cell lineage (KEGG 2019 pathway), immune system (Reactome pathway 2016), and integrin family cell-surface interactions (NCI-Nature Pathway 2016) (Figure 6a–c). In GO analysis, hub genes were significantly enriched in regulation of immune response (biological process 2018), hydrolase activity, hydrolyzing N-glycosyl compounds (molecular function 2018), and clathrin-coated vesicle membrane (cellular component 2018) (Figure 6d–f).

### 3.5. Expression Analysis of Hub Genes in GEO Datasets and IC/BPS Patient Tissues

Finally, the expression of six hub genes was analyzed in GEO datasets and subsequently in IC/BPS patient tissues compared to adjacent normal tissue. For GEO analysis, the expression level of each gene was extracted from respective GEO datasets, presented as boxplots, and subsequently subjected to statistical analysis by a two-tailed unpaired *t*-test. Based on the GEO dataset (GSE11783 and GSE57560) analysis, all the hub genes were significantly overexpressed in IC/BPS patients compared to normal patients (Figure 7a,b). Among the hub genes, *CD38*, *PSMB9*, *CD5*, *ITGAL*, and *KLRB1* were most significantly overexpressed in IC/BPS tissues, while *IL7R* was the least expressed (Figure 7a,b).

Next, we analyzed the mRNA expression of hub genes from our IC/BPS patients and normal tissue samples, which were provided by Konkuk University Medical Center. The clinicopathological characteristics of non-IC/BPS and IC/BPS patients are shown in Table 3. Our RT-qPCR data showed that all six hub genes were upregulated in IC/BPS tissues relative to the control tissues (Figure 8), although there were differences in relative expression among IC/BPS samples. Of importance, our data showed that *PSMB9*, *ITGAL*, and *KLRB1* were significantly overexpressed in IC/BPS samples compared to non-IC/BPS samples, in agreement with GEO data, which indicated that *PSMB9*, *ITGAL*, and *KLRB1* might have the highest potential to be biomarkers of IC/BPS among all hub genes (Figure 8). The expression of *CD38* and *IL7R* also demonstrated a significant difference between non-IC/BPS and IC/BPS patients (Figure 8). In addition, another hub gene, *CD5*, revealed the upregulated mean expression pattern in IC/BPS samples compared to non-IC/BPS samples. However, the difference was not significant (*p* = 0.31), which might be due to the limited number of samples (Figure 8). Therefore, these results suggest that at least five out of six hub genes may be potential diagnostic biomarkers and are likely associated with the etiology of IC/BPS, which deserves extensive study.

## 4. Discussion

The etiology of IC/BPS is poorly understood; however, genetic mechanisms have been suggested to play a role. We propose that genome-based expression profiling can be used in the diagnosis of the Hunner type of IC/BPS in clinical practice. In the present study, we revealed novel diagnostic methods to identify the disease. Thus far, clinicians have only been able to use subjective symptoms and interviews from patients for diagnosis. In addition, surgeons can view only ‘chronic inflammation or cystitis’ in the pathologic report after transurethral resection of Hunner lesions. It could bias the diagnosis of the disease. Instead, clinicians and pathologists could use genetic biomarkers to form a correct diagnosis of IC/BPS. Tools to accurately facilitate this is lacking, and so far, clinically useful biomarkers are still under investigation [19]. Therefore, we propose using genetic biomarkers to improve the accuracy of diagnosis using bioinformatics analysis.

There is no doubt that the key to diagnosis is obtaining a careful history and identifying characteristic symptoms, including urinary bladder pain perception, urinary frequency, and urgency. Pain can be localized to the bladder, or rather in the area that the patient perceives as the bladder, or in the lower abdomen and pelvis. The classic description is a compelling urge upon bladder filling with increasing suprapubic pain, in many instances very severe, that is relieved by voiding, although symptoms soon return. In the last decade, the definition of IC/BPS changed significantly, especially after the exclusion of cystoscopic abnormalities as necessary findings for diagnosis [20]. Misdiagnosis occurred because, although some patients showed signs of inflammation on cystoscopy, the old definition was restrictive and encompassed only one-third of patients with a presumptive diagnosis of IC/BPS [21,22]. Cystoscopy under anesthesia is still the cornerstone for investigation of IC/BPS patients to visualize lesions characteristic of IC/BPS, with particular attention to the presence or absence of Hunner lesions. Typical lesions are located in mucosal areas and are reddened and surrounded by small vessels spreading towards central scars with fibrin deposits or coagulum attached. It is not an actual ulcer, but rather a very vulnerable, inflamed area [23]. However, detection is undoubtedly a matter of attention and training. The cystoscopic diagnosis of Hunner lesions is often difficult or impossible without bladder distension under general anesthesia. The presence of submucosal petechial bleedings, so-called glomerulations, after decompression of the previously distended bladder, has until recently been considered as one of the endoscopic hallmarks of the disease. Recent data have, however, raised concerns over the diagnostic usefulness of this detection. According to previous reports, the value of glomerulations as a reliable diagnostic variable for IC/BPS is doubtful, and it was shown that the cystoscopic appearance of the bladder wall after hydrodistension is not necessarily constant [24].

The primary purpose of obtaining bladder biopsies in IC/BPS is to exclude other causes of bladder pain, including serious diseases, such as carcinoma in situ. Histopathological examination is also of value to diagnose IC/BPS. It includes urothelial vacuolization and detachment, mucosal infiltrate of lymphocytes, plasma cells, and neutrophil and eosinophil granulocytes, and an increase of mast cells in all compartments of the bladder wall [22,25]. However, all outcomes are from laboratories and not clinical settings. Unfortunately, ‘chronic cystitis’ or ‘chronic inflammation’ can alone be written in pathological reports. The prevalence of Hunner lesions varies widely, between 5% and 57%, which raises concerns on whether diagnostics are universally applicable and the rate of misdiagnosis. To date, there are no gold standards of pathologic reports in the diagnosis or detection of IC/BPS, and clinicians have to rule out several symptoms common to these co-morbid diseases (i.e., differential diagnosis) to begin treatment for IC/BPS.

We revealed six hub genes, *CD5*, *CD38*, *ITGAL*, *IL7R*, *KRLB1*, and *PSMB9*, as novel biomarkers through bioinformatic analysis, and our experimental results with Korean patients confirmed that the hub genes could be potential diagnostic biomarkers for IC/BPS with Hunner lesions. Our experimental results showed that, among the six hub genes, in particular, *PSMB9*, *ITGAL*, and *KLRB1,* might have the highest potential to be biomarkers of IC/BPS and *CD38* and *IL7R* were also significantly upregulated in IC/BPS patients with Hunner legions. The expression of *CD5* also appeared to be upregulated in IC/BPS samples, although the difference was not significant which might be due to the limited number of samples. These findings are commonly associated with autoimmune disease. Autoimmunity is one of the proposed causes of IC/BPS. The evidence to support this includes the similarity of sex and age distribution of patients with IC/BPS to those of known autoimmune diseases [26,27]. Autoantibodies, such as those against nuclear and bladder epithelium antigens, have been found in patients with interstitial cystitis, but these are likely to be secondary to the disease. Indirect evidence does support a possible autoimmune nature of interstitial cystitis, such as the strong female preponderance and the clinical association between interstitial cystitis and other known autoimmune diseases within patients and families. The strongest association occurs between interstitial cystitis and Sjögren’s syndrome. Increasing evidence suggests a possible role of autoantibodies to the muscarinic M3 receptor in Sjögren’s syndrome. The M3 receptor is also located on the detrusor muscle cells of the bladder and mediates cholinergic contraction of the urinary bladder and other smooth muscle tissues. Autoantibodies to the M3 receptor might be important in both the early noninflammatory and the late inflammatory features of interstitial cystitis. Other autoimmune conditions, such as fibromyalgia, have also been shown to bear resemblance or be associated with IC/BPS. Fibromyalgia shows antibodies against urothelium, smooth muscle, and connective tissue components of the urinary bladder.

Indeed, the pathophysiology of those conditions is similar to that of IC/BPS [27]. Moreover, numerous evidences suggest that autoimmune mechanisms and T cell infiltration play a pivotal role in the pathogenesis of IC/BPS [28,29]. Our novel findings are similar to those of previous research on relationships between autoimmune disease and IC/BPS. Large-scale, population-based studies have reported that Sjögren’s syndrome and systemic lupus erythematosus patients had a significantly higher risk of IC/BPS [30,31]. These studies suggest that autoimmune disease and IC/BPS probably share the same autoimmune nature. IC/BPS patients may possess many of the clinical and pathological features of autoimmune diseases, including the presence of autoantibodies and chronic inflammation, suggesting that the disease can be a chronic condition of the bladder involving an autoimmune response. The hypothesis is that increased inflammatory cytokines induced by auto-antibodies could induce the activation of mast cells [32,33,34]. As we know, mast cells in the bladder lamina propria of the detrusor muscle are also important mediators of the pathogenesis of IC/BPS [31]. The other candidate genetic biomarkers are also involved with immunologic reactions. Chronic inflammation induced and maintained by immune reactions resulted in the modulation of chemokine receptor expression on B cells and mast cells. The serial reactions might injure the protective barrier of the bladder mucosa, which is the pathology of IC/BPS.

Although promising, the clinical significance of such genetic biomarkers from bioinformatics analysis has not been clearly established. The outcomes presented here provide clues to pathophysiological mechanisms of the disease and information for the design of further studies. For IC/BPS biomarkers to be implemented into clinical practice, findings would have to be replicated in patients with varied clinical presentations, and prospective validation studies would have to be performed. Moreover, future work should investigate the relationship between biomarkers and IC/BPS symptoms using validated questionnaires. This would enable future comparisons between studies, in addition to changes in biomarker levels in response to treatment and symptom improvement. The strengths of this research comprise proposing a new approach to evaluating genes from only those patients who fulfill current criteria for IC/BPS. An additional strength is that our study includes the three largest databases in our search strategy.

## 5. Conclusions

In this study, we performed a bioinformatics analysis using patient-derived tissue genomics data to reveal the molecular signatures of IC/BPS. The identified common DEGs were predominantly enriched in immune and metabolic signaling pathways. Six key hub genes were identified using a PPI network analysis of DEGs from three GEO datasets, which were significantly upregulated and interacted more between DEGs. In addition, the datasets samples included urine sediments and bladder tissues. These hub genes might be considered candidate biomarkers that can diagnose the disease in an invasive or non-invasive way. In addition, these hub genes may also be regulated by various transcription factors, which may be related to IC/BPS. Our RT-qPCR data also confirmed the upregulation of hub genes in our IC/BPS patient tissues compared to normal tissues. As a result, at least five out of six hub genes were finally selected. Thus, these five hub genes might play a potential role in IC/BPS development and serve as diagnostic biomarkers of IC/BPS, which deserves further detailed investigation.

## Figures and Tables

**Figure 1 jcm-09-01935-f001:**
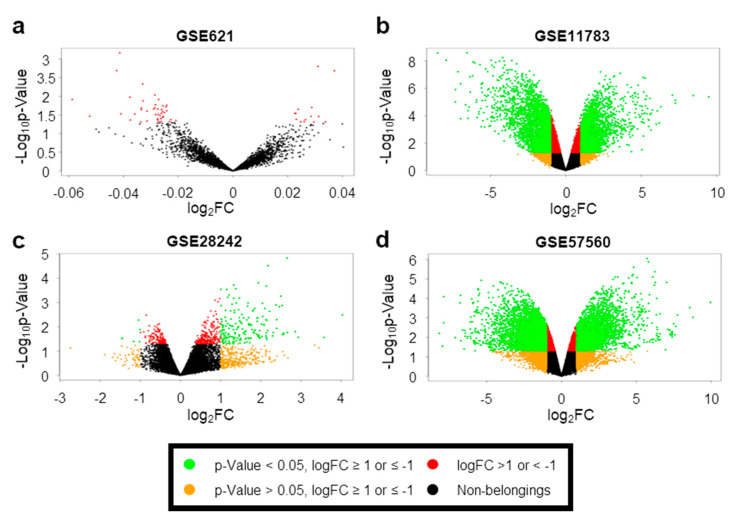
Analysis of significant genes in interstitial cystitis/bladder pain syndrome (IC/BPS). (**a**–**d**) Volcano plots show the transcript expression between normal and IC/BPS patients. Color indicates each condition (green: *p*-value < 0.05, logFC ≥ 1 or ≤ −1; orange: *p*-value > 0.05, logFC ≥ 1 or ≤ −1; red: logFC >1 or < −1; black: non-significant).

**Figure 2 jcm-09-01935-f002:**
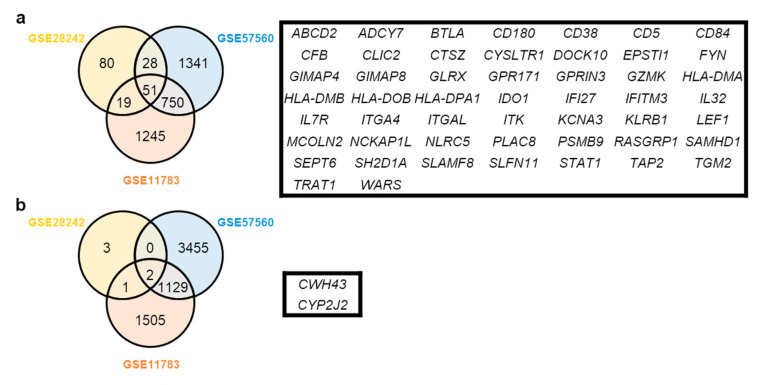
Analysis of common positive and negative genes in IC/BPS. The intersections are common significantly expressed genes from 3 datasets of the Gene Expression Omnibus (GEO) database. There are positive gene groups (**a**) and negative groups (**b**).

**Figure 3 jcm-09-01935-f003:**
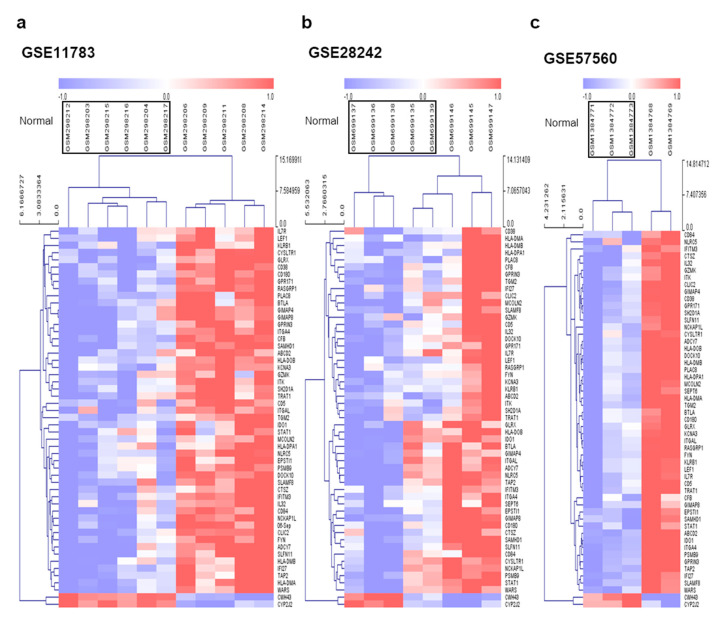
Expression of common differentially expressed genes (DEGs) in IC/BPS datasets. The heatmap represents the expression and tendency of intersected genes in each dataset. These data were analyzed by hierarchical clustering using the MeV software. The color of the pattern depends on the level of gene expression with a threshold of −1 < z-score < 1, where the high and low values outside of the criteria were assigned the highest and lowest values. (**a**) GSE11783, (**b**) GSE28242, and (**c**) GSE57560 are shown.

**Figure 4 jcm-09-01935-f004:**
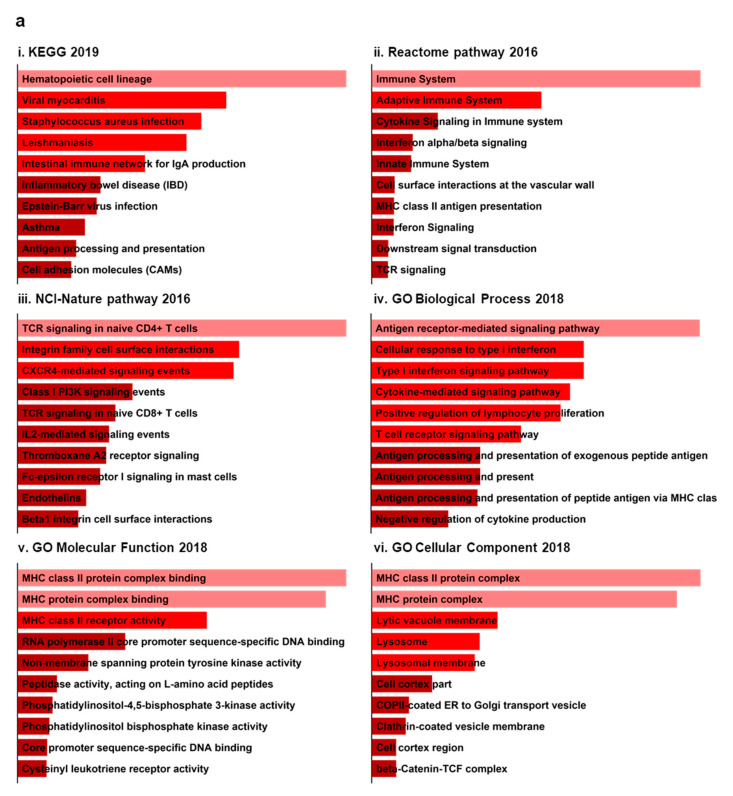
Putative pathways and Gene Ontology (GO) of common DEGs in IC/BPS. The bar graphs from the Enrichr platform obtained the Kyoto Encyclopedia of Genes and Genomes (KEGG), Reactome pathway, National Cancer Institute (NCI)-Nature Pathway, and Gene Ontology (*p*-value ≤ 0.05, top 10 pathways of significance). There are red bars for upregulated genes (**a**) and blue bars for downregulated genes (**b**). Red bars indicate upregulated common DEG groups. Conversely, blue bars represent downregulated common DEG groups. The darker bar color indicates a less significant *p*-value.

**Figure 5 jcm-09-01935-f005:**
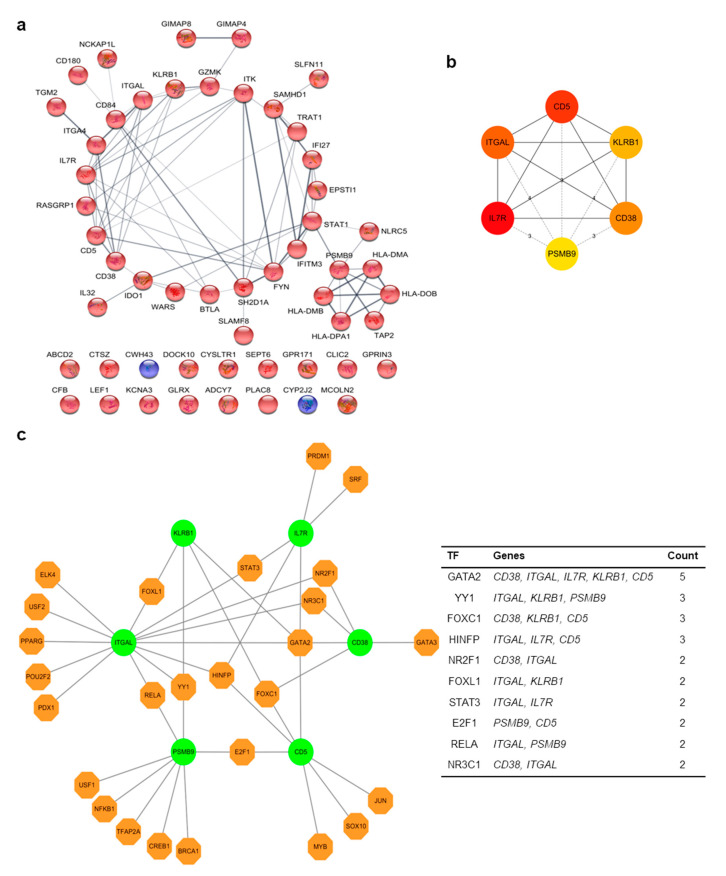
Analysis of hub genes and their putative pathways. (**a**) STRING predicted functional protein interactions of significant genes, both positive and negative. The edge indicates the interaction degree between nodes. The color of nodes means upregulated (red) or downregulated (blue) genes in IC/BPS patients compared to control. (**b**) In this map, hub genes and the top six protein interactions are shown. They were obtained using cytoHubba from Cytoscape. The node color is redder depending on significant interactions. (**c**) Hub gene–transcriptional factor (TF) interaction was analyzed using JASPAR database through NetworkAnalyst. The edge indicates that the hub gene is associated with putative TFs. The color of the node is different depending on the roles; green is the hub gene and orange is the TF.

**Figure 6 jcm-09-01935-f006:**
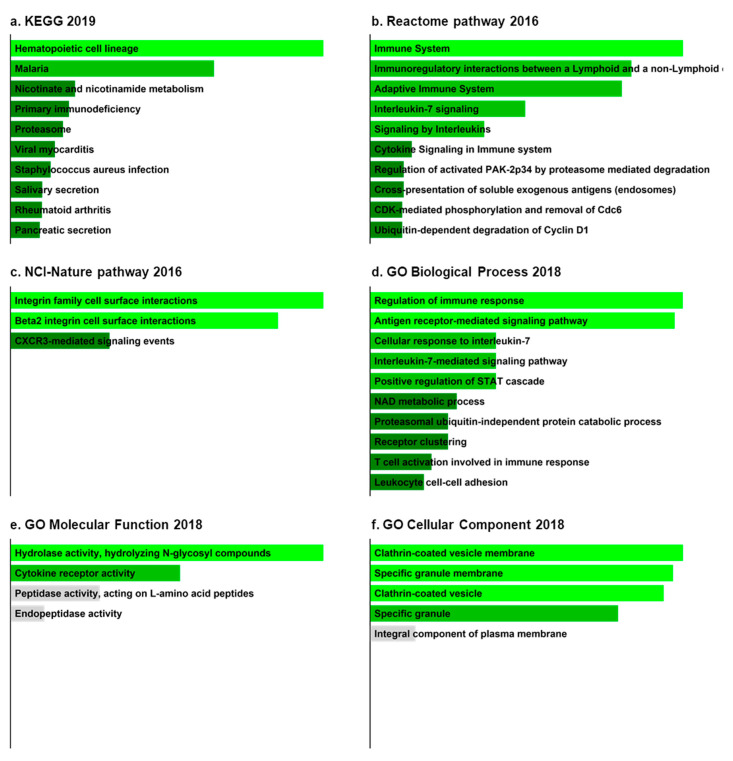
Putative pathways and GO of hub genes in IC/BPS. (**a**–**f**) Bar graphs indicate the putative pathway acquired from various pathway databases. These results are ranked by *p*-value. Gray bars denote *p*-value > 0.05, and the darker colors and the shorter bars have higher *p*-values.

**Figure 7 jcm-09-01935-f007:**
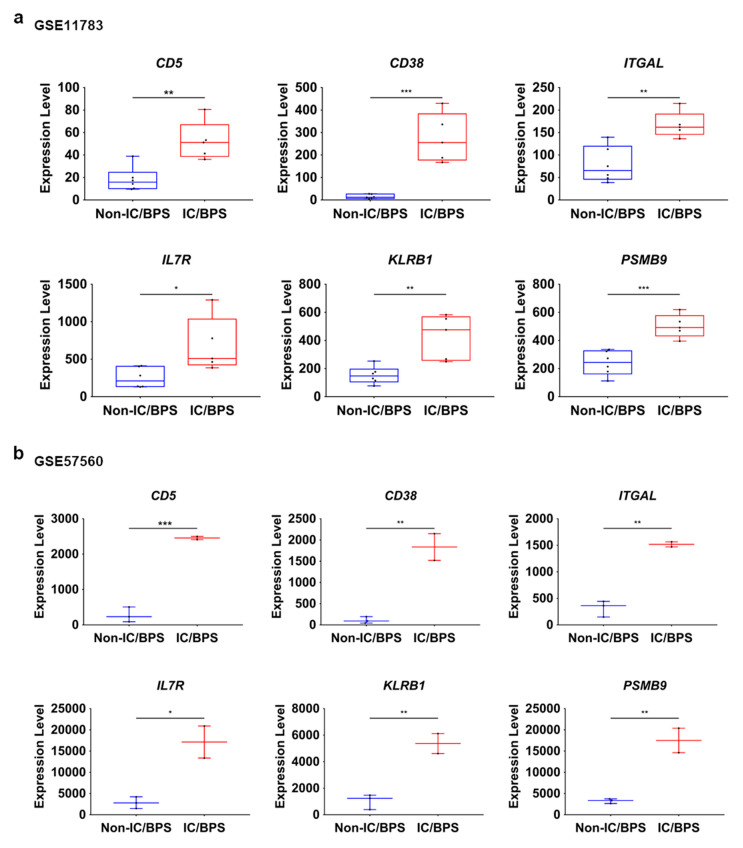
Expression of hub genes in IC/BPS datasets. Box plot data were acquired from GEO datasets, which included (**a**) GSE11783 and (**b**) GSE57560. Comparison of expression of 6 hub genes between IC with Hunner lesiosn (HL) and control. These box plots were made, and statistical analysis performed, using GraphPad prism. The *p*-value was analyzed using a two-tailed unpaired *t*-test. Asterisk symbols indicate * *p* < 0.05, ** *p* < 0.01, and *** *p* < 0.001.

**Figure 8 jcm-09-01935-f008:**
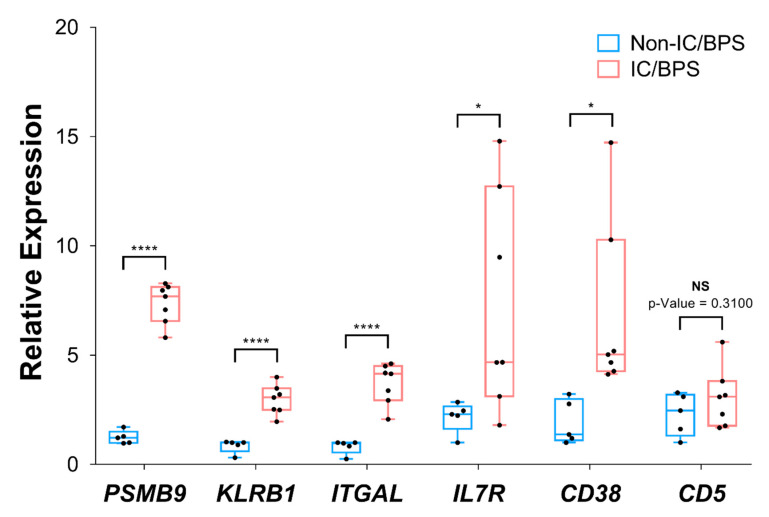
Relative mRNA expression of six hub genes was analyzed by RT-qPCR using non-IC/BPS (*n* = 5) and IC/BPS (*n* = 7) tissue samples. Expression of genes was normalized with *GAPDH*, a housekeeping gene, expression. * *p* < 0.05, **** *p* < 0.0001, NS: non-significant.

**Table 1 jcm-09-01935-t001:** Inclusion and exclusion criteria of the study population.

**Inclusion Criteria**
Persistent pelvic pain related to the bladder
Pain worsening with bladder filling
Duration of symptoms longer than 6 weeks
Presence of Hunner lesions and glomerulation in cystoscopic finding
**Exclusion Criteria**
Untreated UTI, STD, or urinary stone
Any defined neurological disease affecting the pelvic nerve
History of endometriosis
Urethral diverticulum
Suspicious DRE or PSA ≥2.5 ng/mL

Abbreviations: UTI: urinary tract infection; STD: sexually transmitted disease; DRE: digital rectal examination; PSA: prostate specific antigen.

**Table 2 jcm-09-01935-t002:** List of primers used in this study.

Gene Symbol	Direction	Sequence
*GADPH*	Forward	GTC TCC TCT GAC TTC AAC AGC G
	Reverse	ACC ACC CTG TTG CTG TAG CCA A
*CD38*	Forward	CAG ACT GGA GAA AGG ACT GC
	Reverse	TTT ACT GCG GGA TCC ATT GAG
*CD5*	Forward	CGA GTT CTT GCC CTC CTT TGC T
	Reverse	TCC TGG CTG AAG AGC TGT CAC A
*IL7R*	Forward	ATC GCA GCA CTC ACT GAC CTG T
	Reverse	TCA GGC ACT TTA CCT CCA CGA G
*ITGAL*	Forward	CTG CTT TTG CCA GCC TCT CTG T
	Reverse	GCT CAC AGG TAT CTG GCT ATG G
*KLRB1*	Forward	GTT CCA CCA AAG AAT CCA GCC TG
	Reverse	AAG AGC CGT TTA TCC ACT TCC AG
*PSMB9*	Forward	CGA GAG GAC TTG TCT GCA CAT C
	Reverse	CAC CAA TGG CAA AAG GCT GTC G

**Table 3 jcm-09-01935-t003:** Clinicopathological characteristics of the study population.

No.	Sex	Age	VAS Score	ICSI Score	ICPI Score	PUF Symptom Score	PUF Bother Score	Symptom Duration (months)
Non-IC/BPS 1	M	73	0	0	0	0	0	0
Non-IC/BPS 2	M	73	0	0	0	0	0	0
Non-IC/BPS 3	M	73	0	0	0	0	0	0
Non-IC/BPS 4	M	72	0	0	0	0	0	0
Non-IC/BPS 5	M	72	0	0	0	0	0	0
IC/BPS 1	M	26	8	18	18	22	12	24
IC/BPS 2	F	47	10	20	20	22	15	30
IC/BPS 3	M	57	8	18	20	20	12	38
IC/BPS 4	F	73	10	20	20	20	15	36
IC/BPS 5	F	74	10	18	20	22	15	26
IC/BPS 6	F	72	8	16	20	20	15	30
IC/BPS 7	F	70	8	20	20	28	15	30

VAS: visual analog scale; PUF: pain urgency frequency questionnaire; ICSI: interstitial cystitis symptom index; ICPI: interstitial cystitis problem index.

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
