# Peer review of "Bioinformatics Approach for Identifying Novel Biomarkers and Their Signaling Pathways Involved in Interstitial Cystitis/Bladder Pain Syndrome with Hunner Lesion"

_jcm, 2020, doi:10.3390/jcm9061935_

Round 1
Reviewer 1 Report
This manuscript can be accepted now with the changes made by the author.
Author Response
We would like to thank you and the reviewers for the useful comments.
Responses to the Comments of Reviewer 1
Reviewer’s Comment: This manuscript can be accepted now with the changes made by the author.
Our response: We like to thank you very much for your time to read and comment on our manuscript.
Reviewer 2 Report
The manuscript reports the bioinformatic analysis of existing gene expression omnibus (GEO) datasets from interstitial cystitis/bladder pain (IC/BPS) syndrome patients with the aim to validate novel biomarkers. The research is significant, as diagnosis of IC/BPS is challenging. The condition is usually diagnosed through symptoms (e.g. pain or discomfort associated with bladder filling) and elimination of other potential causes (UTI, malignancies).
The authors have presented a well written and interesting study. However, there are several issues that need to be addressed.
Key issues
1) The title should indicate the study examined IC/BPS patients with Hunner's ulcers. Recent consensus is that Hunner's and non-Hunner's IC/BPS patients should be considered as separate pathologies (Intl. J. Urol, 26(s1), pg26-34, 2019). This is also supported by recent RNA-seq data indicating differences in gene expression profiles between these IC/BPS groups (J. Urol. 202(2), pg290-30, 2019).
2) Were only mucosal tissues sampled for RNA extraction? If so this should be indicated in the methods.
3) The results should include a table of the patient demographic information (e.g. number of patients of each sex, age range, average symptom scores).
4) The differences in the biosample types used in the GEO datasets needs to be discussed. Specifically, GSE28242 which examined urine sediment from IC patients. This may reflect the smaller number of overlapping DEG in this dataset and raises the question whether this affects the DEGs identified in this study.
Minor issues
Results, line 151. GES28242 should be GSE.
Discussion, line 365. Why have the words "six" and "CD5" been crossed out? The rest of the manuscript indicates there are six key genes.
Author Response
Please check the attached word file (200601_JCM revision-Fn3).

Round 2
Reviewer 2 Report
The amended manuscript is acceptable and ready for publication.